# DRFormer: A Benchmark Model for RNA Sequence Downstream Tasks

**DOI:** 10.3390/genes16030284

**Published:** 2025-02-26

**Authors:** Jianqi Fu, Haohao Li, Yanlei Kang, Hancan Zhu, Tiren Huang, Zhong Li

**Affiliations:** 1School of Information Engineering, Huzhou University, Huzhou 313000, China02959@zjhu.edu.cn (Y.K.); 2College of Science, Zhejiang Sci-Tech University, Hangzhou 310018, China; hhl820@zstu.edu.cn (H.L.); htiren@zstu.edu.cn (T.H.); 3School of Mathematics, Physics and Information, Shaoxing University, Shaoxing 312000, China; hancan-zhu@yeah.net

**Keywords:** RNA, RSS, RBP, sequence classification, multimodal

## Abstract

**Background/Objectives:** RNA research is critical for understanding gene regulation, disease mechanisms, and therapeutic development. Constructing effective RNA benchmark models for accurate downstream analysis has become a significant research challenge. The objective of this study is to propose a robust benchmark model, DRFormer, for RNA sequence downstream tasks. **Methods:** The DRFormer model utilizes RNA sequences to construct novel vision features based on secondary structure and sequence distance. These features are pre-trained using the SWIN model to develop a SWIN-RNA submodel. This submodel is then integrated with an RNA sequence model to construct a multimodal model for downstream analysis. **Results**: We conducted experiments on various RNA downstream tasks. In the sequence classification task, the MCC reached 94.4%, surpassing the state-of-the-art RNAErnie model by 1.2%. In the protein–RNA interaction prediction, DRFormer achieved an MCC of 0.492, outperforming advanced models like BERT-RBP and PrismNet. In RNA secondary structure prediction, the F1 score was 0.690, exceeding the widely used SPOT-RNA model by 1%. Additionally, generalization experiments on DNA tasks yielded satisfactory results. **Conclusions:** DRFormer is the first RNA sequence downstream analysis model that leverages structural features to construct a vision model and integrates sequence and vision models in a multimodal manner. This approach yields excellent prediction and analysis results, making it a valuable contribution to RNA research.

## 1. Introduction

RNA is a pivotal molecule in the central dogma of molecular biology. As a multi-functional macromolecule and genetic information carrier, it describes the flow of genetic information from DNA to RNA and ultimately to protein [1]. RNA plays a crucial role in various cellular processes, such as gene expression, regulation, and catalysis, and has become an essential regulatory factor influencing life processes [2]. Given RNA’s significance in biological systems, there is an increasing demand for efficient and accurate methods to analyze RNA sequences.

Traditionally, RNA sequences are analyzed through experimental techniques such as RNA sequencing and microarray analysis [3]. For instance, cDNA preparation, sequencing and read mapping on the genome and across splices [4], and the widely adopted dUTP method [5]. However, these methods are often time-consuming and expensive. In recent years, computational approaches based on machine learning have been employed to analyze RNA sequences. For example, Adam McDermaid et al. proposed GeneQC [6], a machine learning tool, to accurately estimate the reliability of each gene expression level derived from RNA-Seq datasets. With the advent of artificial intelligence, deep learning methods have also been utilized for RNA sequence analysis and various downstream tasks. These methods have been widely applied in specific tasks, for example, protein–RNA interaction prediction by DeepBind [7] and RNA secondary structure prediction by MXFold2 [8].

Recently, pre-trained language models have achieved remarkable success in various natural language processing (NLP) tasks, such as text classification, question answering, and language translation. Researchers have increasingly applied the pre-trained language models like BERT to simulate and analyze the large-scale biological sequences. For instance, in RNAErnie proposed by Wang et al. [9], RNA motifs are incorporated as biological priors. This approach introduces the motif-level random masking and sub-sequence-level masking in language modeling to enhance the pre-training. Additionally, RNA types are added as the stop words during pre-training, which can effectively separate the word embeddings among different RNA types, leading to the improved representational capacity. In DNABERT proposed by Ji et al. [10], biological sequences are tokenized using k-mers, followed by BERT-based pre-training, which can effectively capture sequence information. Followingly, Zhou et al. [11] introduced an advanced approach, DNABERT-2, to replace the k-mer tokenization with Byte Pair Encoding (BPE), overcoming the limitations of k-mer tokenization. This enhanced tokenization strategy, combined with BERT-based pre-training, significantly improves the model’s learning capability.

Currently, most benchmark models use sequences as input, relying solely on the unimodal representation of sequences to predict RNA structures and perform various downstream tasks. However, as we know for RNA, the biological sequence determines its structure, thereby affecting its function. Consequently, relying solely on the sequence-based unimodal benchmark models may not yield the accurate results for downstream analysis. In the traditional deep learning field, structural information can be used to complement sequence information for specific tasks. For instance, the protein–RNA interaction prediction, PrismNet, proposed by Xu et al. [12], incorporates icSHAPE data (which represents single-stranded and double-stranded structural information of sequences) as a supplement, achieving significant prediction improvement. Therefore, it is crucial to explore multimodal approaches that integrate sequence and structural information to construct benchmark models, thereby enhancing the analysis of downstream tasks.

In terms of structural information, we notice that there is the emergence of some vision-based deep learning models for RNA downstream analysis. For example, in the RNA secondary structure prediction (RSS), UFold [13] is proposed to extract features based on RNA secondary structure information, while RNAformer [14] is designed, which uses the word embeddings to extract features. In this work, we firstly construct novel structural vision features based on RNA secondary structure matching rules. Using these features, we pre-train a vision-based benchmark model SWIN-RNA with the SWIN transformer [15], and then integrate the vision benchmark model with a sequence benchmark model to form a multimodal model DRFormer. We conduct extensive downstream analyses by this multimodal model, including the sequence classification, protein–RNA interaction prediction, and RNA secondary structure prediction. Additionally, we perform the generalization experiment on DNA downstream tasks. All experiments yield satisfactory results, demonstrating the versatility and effectiveness of DRFormer as a general-purpose solution.

## 2. Materials and Methods

### 2.1. Datasets

We utilize the NONCODEV6 database [16] as our pre-training dataset, which contains a total of 0.3 billion bases. For the RNA secondary structure prediction, we employ several datasets, including bpRNA-1m by Danaee et al. [17], ArchiveII by Sloma et al. [18], and TrainSetA, TestSetA, and TestSetB provided by Rivas et al. [19]. The bpRNA-1m dataset is one of the most comprehensive RNA structure datasets, comprising 102,318 sequences from 2588 families. Following the preprocessing approach of MXFold2 [8], we use CD-HIT to remove redundant sequences from bpRNA-1m and generate a training set (TR0) and a test set (TS0). The ArchiveII dataset is one of the most widely used benchmark datasets for RNA secondary structure prediction, which includes 3975 sequences from 10 families. We perform the five-fold cross-validation on ArchiveII and conduct the secondary structure prediction experiment on the TS0 and ArchiveII datasets. Furthermore, we perform the feature ablation studies on the TS0, TestSetA, and TestSetB datasets to validate the contributions of different features on our model’s performance.

For the protein–RNA interaction prediction, we utilize the CLIP dataset [20], which is one of comprehensive datasets available. It contains data for 168 RNA binding proteins (RBPs). For each RBP, the positive samples are the top 5000 signal binding sites with an icSHAPE score coverage of at least 40%, while the negative samples are 10,000 randomly selected RNA regions from the transcriptome, also with an icSHAPE score coverage of at least 40%, and simultaneously avoiding the corresponding RBP binding regions. For the pre-training, 10% of non-overlapping sequences from each RBP are extracted. The remaining data are split into training and testing sets, with 80% used for training and 20% for testing. This setup ensures the robust evaluation of the model’s performance on the protein–RNA interaction prediction.

For the RNA sequence classification, we utilize the nRC dataset [21], which comprises ncRNA sequences selected from the 12th edition of the Rfam database. The nRC dataset is a balanced collection of sequences, with each of the 13 categories containing 20% non-redundant samples. It includes 6320 training sequences and 2600 test sequences, all labeled into 13 distinct classes. This dataset serves as a standard benchmark for evaluating the RNA sequence classification models.

In experiments on different DNA generalization tasks, we use the GUE dataset [11], which is currently used in the DNA field for benchmark model comparison. It contains 28 datasets with seven genome sequence analysis problems, and sequence length ranges from 70 to 1000. The dataset contains tasks for various species, including humans, fungi, viruses, and yeast, and it is divided into training, validation, and test datasets for each individual task.

### 2.2. Features

In the RNA sequence benchmark model, we still use the k-mer method to extract sequence features. Namely, *k* consecutive bases in the RNA sequence are merged into a token, and then the word embedding is used to obtain the feature information. For a sequence of length L, L−k+1 tokens can be obtained, and then two tokens CLS and SEP are added, finally forming a total of L−k+3 tokens. After the word embedding, the feature size is $\left(L-k+3\right)\times D^{t}$, where $D^{t}$ represents the dimension of word embedding.

In terms of structural vision models, we introduce the longest possible secondary structure, sequence repeat structure, base missing structure, and spatial distance feature based on existing RNA coding matrix, Manhattan distance feature, and base pairing signature. To fully explore the structural and distance features, we combine them to propose a more comprehensive 24-dimensional feature and the feature size of each dimension is L×L, where *L* is the sequence length. Therefore, it finally forms a structural feature with a total size of 24×L×L.

#### 2.2.1. Structural Signature (SS)

Longest possible secondary structure (LPSS) refers to the longest possible structure that meets the secondary structure condition. This feature contains all possible secondary structure information and serves as an important feature supplement for possible secondary structures. Specifically, we calculate the longest fragment length in the sequence that the current base fragment can be matched by the dynamic programming, namely, compute the maximum value of the pairing in each base fragment according to three pairing rules [22], finally obtaining a feature of size *L* × *L*. For detailed calculations, see Section A.1.

Sequence repeat structure (SRS) refers to the possibility that a base may be paired with multiple different bases in the longest possible secondary structure. Therefore, we count the number of bases that can be paired, and use this information to calculate the importance of the corresponding base. Similarly, we use the dynamic programming to calculate the number of bases that can be repeatedly paired in the sequence interval, and obtain a total of five feature maps with a size of 5 × *L* × *L*. For detailed calculations, see Section A.2.

Base missing structure (BMS) refers to the situation where a base may be missing in the longest possible secondary structure. The situation of this base can be reflected by the number of other paired bases. For a given base, we calculate the number of bases that are paired according to the first pairing rule from three pairing rules, and finally obtain a total of five feature maps with a size of 5 × *L* × *L*. For specific calculations, see Section A.3.

RNA coding matrix (RCM) [23] refers to the RNA matrix representation based on the RNA sequence pairing feature proposed in CDPfold, which is also a supplement to the possible existence of secondary structure. We calculate the pairing probability between each base and other bases in a sequence according to three pairing rules, and obtain a feature map of size *L* × *L*.

#### 2.2.2. Distance Feature (DIS)

Spatial distance feature considers the relationship between bases and their positions in the sequence, we add the position information for spatial distance calculation. We binarize the base position and add the corresponding base’s one-hot encoding at the end of binary encoding. Then we calculate the Euclidean distance between two bases to obtain the spatial distance feature, and obtain a feature map of size *L* × *L*. For the detailed calculation, see Section A.4.

Manhattan distance feature [24] is usually regarded as an important reference in the RNA analysis model, so Manhattan distance is used here as a distance feature. After the bases are encoded by one-hot encoding, the Manhattan distance between two bases is calculated, and a feature map of size *L* × *L* is obtained.

#### 2.2.3. Base Pairing Signature (BPS)

The base pairing feature adopts the feature proposed by Fu [13]. The base is converted into a one-hot encoding to obtain a size of 4 × *L*, and then the feature is converted to a 16 × *L* × *L* tensor through Kronecker product, which can be understood as an image matrix of *L* × *L* with 16 color channels. These 16 channels represent the base pairing situation for the base set {A, U, C, G}. Note that there is the redundancy in these base pairing matrices, for example, A-U and U-A pairings are regarded as the same since the corresponding pairing matrix is equal to its transposed matrix. We delete 6 redundant channels and finally obtain a 10 × *L* × *L* feature tensor as the base pairing signature.

### 2.3. Model

#### 2.3.1. Overall Architecture

In our proposed DRFormer model, we firstly design a large module based on structural vision; that is, using the constructed 24-dimensional structural features and the SWIN pre-training method designed by GreenMIM [25] for pre-training, to generate a vision model that can effectively extract RNA structure information. At the same time, based on the multimodal strategy, we adopt a method of fusing sequence and vision information to build a multimodal module that can better mine sequence and structural vision features, as shown on the left side of Figure 1. In terms of the sequence module and structural vision module, we apply DNABERT (3-mer) and SWIN-RNA models, respectively.

Based on the multimodal module, we design two downstream networks to analyze various downstream tasks. The first task is the RNA secondary structure prediction. SWIN-RNA is firstly used for downsampling, and then we design an upconvolution operation for upsampling. It follows the U-Net structure of encoder–decoder and keeps skip-connections in the middle of U-Net, as shown in the lower right of Figure 1. The second task is the classification of RNA sequences and the prediction of protein–RNA interactions. The fusion block of self-attention and cross-attention is used to fuse vision (structure) and text (sequence) information, and then a fusion layer is applied to obtain the classification result, as shown in the upper right of Figure 1.

#### 2.3.2. Benchmark Model

SWIN-RNA: SWIN is a commonly used Transformer-based model in the field of image processing. It contains SWB (Two Successive SWIN Transformer Blocks) based on W-MSA (window-based multi-head self-attention) and SW-MSA (shifted window based multi-head self-attention), which can be used to extract vision features as well. We construct the structural signature, distance feature, and base pairing signature in Section 2.2 to obtain a 24×L×L structural vision feature x. Then this feature is downsampled by the PE (Patch Embed) block, and the shallow feature extraction is performed through two SWB blocks. After *n* PM (Patch Merging) and *m* SWB fusion blocks, a $D^{l}\times \frac{L}{16}\times \frac{L}{16}$ structural vision embedding is obtained as(1)$$V^{l}=SWB\left(PM\left(\;SWB\left(PE\left(x\right)\right)\right)\right)$$

DNABERT-3: DNABERT obtains the corresponding token by performing *k*-mer on the sequence, and then uses the BERT model for pre-training. DNABERT is an easy-to-use pre-trained sequence model with an open source, which performs well on the related RNA tasks, such as BERT-RBP. For a sequence of length *L*, we execute a 3-mer operation to obtain the corresponding word segmentation result, add special word segmentations such as [CLS], [ENP], and finally do the embedding to obtain a size of $L\times D^{t}$. Through a 12-layer Transformer encoder block, we obtain the corresponding sequence text embedding as(2)$$T^{l}=Embedding\left(Encoder\;block\left(x\right)\right)$$

#### 2.3.3. Multimodal Fusion

Assume that the output of the sequence module is the text embedding ($T^{l}$) and the output of the structural module is the vision embedding ($V^{l}$). The constructed multimodal model is divided into two parts: the first part is the self-attention module, which uses the self-attention mechanism to better focus on the features of the corresponding modality; and the second part is the cross-attention module, which applies the cross-attention to fuse the features of two modalities.

In the self-attention (SA) module, the original information is mapped to *Q*, *K*, *V*, and the corresponding encoded information is calculated by $Softmax\left(\frac{QK^{T}}{sqrt\left(d_{k}\right)}\right)V$. Self- attention is used to obtain the self-representation of the information on $T^{l}$ and $V^{l}$.

In the cross-attention module, cross-attention (CA) is introduced to obtain information from another modality for fusion. Cross-attention maps $T^{l}$ to *Q* in the attention mechanism, maps $V^{l}$ to the corresponding *K* and *V*, and then obtains $T^{l}$’s attention to $V^{l}$ through $softmax\left(\frac{QK^{T}}{sqrt\left(d_{k}\right)}\right)V$. We similarly do the attention operation of $V^{l}$ to $T^{l}$.(3)$$Fusion\left(T^{l}\right)=SeLU\left(LayerNorm\left(Linear\left(SA\left(T^{l}\right)+CA\left(T^{l}\right)\right)\right)\right)\;Fusion\left(V^{l}\right)=SeLU\left(LayerNorm\left(Linear\left(SA\left(V^{l}\right)+CA\left(V^{l}\right)\right)\right)\right)$$
where SeLU represents the activation function, and(4)$$CA\left(T^{l}\right)=Softmax\left(\frac{Q\left(T^{l}\right)K{\left(V^{l}\right)}^{T}}{sqrt\left(d_{k}\right)}\right)V\left(V^{l}\right),\;CA\left(V^{l}\right)=Softmax\left(\frac{Q\left(V^{l}\right)K{\left(T^{l}\right)}^{T}}{sqrt\left(d_{k}\right)}\right)V\left(T^{l}\right)\;SA\left(T^{l}\right)=Softmax\left(\frac{Q\left(T^{l}\right)K{\left(T^{l}\right)}^{T}}{sqrt\left(d_{k}\right)}\right)V\left(T^{l}\right),\;SA\left(V^{l}\right)=Softmax\left(\frac{Q\left(V^{l}\right)K{\left(V^{l}\right)}^{T}}{sqrt\left(d_{k}\right)}\right)V\left(V^{l}\right)$$

Therefore, the multimodal part is constructed through a fusion block; that is, self-attention and cross-attention are performed on $T^{l}$ to obtain the corresponding $T^{l_{S}}$ and $T^{l_{C}}$, which are connected and passed through a linear layer and a normalization layer. After that, the activation function is implemented to obtain the corresponding fusion result of $T^{l}$. We do the same operation for $V^{l}$. After obtaining $T^{l}$ and $V^{l}$, they can be used for subsequent RNA sequence downstream task analysis.

#### 2.3.4. Downstream Task Network

The classification downstream network can be used on various classification tasks, such as protein–RNA interaction prediction (binary classification) and RNA sequence classification (multi-classification). This network obtains the corresponding classification results through a classification layer after fusing sequence and structural information through multimodality. In the model training, the binary cross-entropy loss is set as the loss function (multi-class cross-entropy loss corresponds to the multi-classification). The main framework is implemented as(5)$$Classifier\left(\frac{1}{C}\overset{C}{\underset{i=1}{\sum }}\left(cat{\left[Fusion\left(T^{l}\right),\;Fusion\left(V^{l}\right)\right]}_{i}\right)\right)$$
where *cat* represents the concatenating of Fusion ($T^{l}$) and Fusion ($V^{l}$) in the channel dimension, *C* is the overall length of text and vision information, which is set to $C=L+\frac{L}{16}$.

RNA secondary structure downstream network (binary classification) can be used on RNA secondary structure prediction. The network acquires a deep extracted feature by cross-attention through text and vision information, and finally obtains the probability matrix for classification through continuous upsampling and jump connection:(6)$$x^{l}=\mathrm{SWINRNA}{\left(x^{l-1}\right)}^{l}\;y^{p}=\mathrm{CA}\left(x^{p}\right)\;y^{l}=conv\left(\mathrm{up}{\left(y^{l+1}\right)}^{l}+x^{l}\right)$$
where *p* is the layer that has undergone the cross-attention (CA), in which no downsampling or upsampling is performed, and only a feature transformation is implemented; *x* represents the change of the downsampled feature; and *y* represents the change of the upsampled feature. For the loss function, a binary cross-entropy loss is used in the training process.

#### 2.3.5. Hyperparameter Settings

In our DRFormer, we use the 3-mer for feature extraction and the dimension of the word embedding is set to 768 in the sequence module. In the vision module, we use a 4-stage design in which the number of SWBs in each stage is 2, 2, 6, and 2, respectively. At the same time, the feature dimension size in the vision module is also 768. In the multimodal fusion module, the number of fusion blocks is 2. For other main parameters, a dropout of 0.1 is set and the AdamW optimizer is used for optimization. A learning rate of 3 × 10^−5^ and weight_decay = 0.05 are set for the RSS task, and a learning rate of 1 × 10^−4^ and weight_decay = 0.05 are used for sequence classification and protein–RNA interaction prediction tasks.

#### 2.3.6. Computational Complexity of Our Model

The computational complexity of our model is divided into three parts, corresponding to sequence module, the vision module, and the final multimodal fusion block, respectively. For the sequence module, we use the DNABERT-3 model, which contains a 12-layer Transformer Encoder. For each layer of the Transformer Encoder, the time complexity is $O\left(4LD^{2}+2L^{2}D\right)$, where *L* represents the length of the sequence and *D* represents the size of the feature dimension.

For the vision module, we use the SWIN-RNA model, which has a computational complexity of $O\left(4HWD^{2}+2M^{2}HWD\right)$, where *M* represents the size of the window, *D* represents the size of the dimension, and *H* and *W* represent the length and width of the image respectively.

In the multimodal fusion block, we implement the self-attention and cross-attention process. The complexity of self-attention after the output of the sequence module is $O\left(4LD^{2}+2L^{2}D\right)$, and the complexity of cross-attention after the output of the sequence module is $O\left(LD^{2}+3L_{v}D^{2}+2LL_{v}D\right)$. The complexity of self-attention after the output of the vision module is $O\left(4L_{v}D^{2}+2L_{v}^{2}D\right)$, and the complexity of cross-attention after the output of the vision module is $O\left(L_{v}D^{2}+3LD^{2}+2L_{v}LD\right)$, where $L_{v}$ is equal to *L*/16. Therefore, the complexity of a complete multimodal fusion block is $O\left(8LD^{2}+8L_{v}D^{2}+2L^{2}D+2L_{v}^{2}D+4LL_{v}D\right)$.

## 3. Results

### 3.1. Evaluation Metrics

Taking into account the imbalance of data distribution and the difference of tasks, we employ different evaluation indicators for various tasks. In the RSS task, we apply F1 as the main indicator, and Sensitivity (Sen) and Positive Predictive Value (PPV) as auxiliary indicators. Here, the F1 value is the harmonic mean of precision and recall, which is used to comprehensively evaluate the performance of classification model. PPV, Sen, and F1 are calculated as follows:(7)$$PPV=\frac{TP}{TP+FP},\;Sen=\frac{TP}{TP+FN},\;F1=\frac{2\times PPV\times Sen}{PPV+Sen}$$
where TP represents the true positive examples, FP represents the false positive examples, and FN represents the false negative examples.

Other involved evaluation indicators include MCC and ACC. MCC is an indicator used to measure the model performance in a binary classification problem, and its value ranges from −1 to 1. ACC (Accuracy) is the ratio of correctly classified samples to the total number of samples.(8)$$MCC=\frac{\left(TP\times TN-FP\times FN\right)}{sqrt\left(\left(TP+FP\right)\times \left(TP+FN\right)\times \left(TN+FP\right)\times \left(TN+FN\right)\right)}$$(9)$$ACC=\frac{\left(TP+TN\right)}{\left(TP+TN+FP+FN\right)}\;$$

In addition, we use AUPRC curves (precision–recall curve) and AUROC (receiver operating characteristic curve) to evaluate the performance of classification models. In the protein–RNA interaction prediction, RNA sequence classification, and DNA downstream tasks, MCC is referred as the primary metric and AUC as the secondary metric for evaluation.

### 3.2. Feature Ablation

We first conduct an ablation experiments on the selected features, and confirm the effectiveness of these features by comparing them on the RSS task. The features include the base pairing feature (BPS), the structural feature (SS), and the distance feature (DIS). We first compare the base pairing feature and structural feature separately, then combine the base pairing feature with structural feature for comparison, and finally add the distance feature for comparison. We compare the results on the TS0_112 dataset, which are shown in Table 1. The results of F1 are 0.587 and 0.613 by the base pairing feature and structural feature, respectively. When the base pairing and structural features are combined together, the F1 changes to 0.631. After adding the distance feature, the F1 reaches 0.639, which is 8.9% higher than using only the base pairing feature and 4.2% higher than using only the structural feature.

Since the structural features include four separate features, namely the longest possible secondary structure (LPSS), RNA coding matrix (RCM), sequence repeat structure (SRS), and base missing structure (BMS), we use the base pairing signature (BPS) as the baseline feature, and then add the corresponding structural features for comparison. The comparison results on the TS0_112 dataset are shown in Table 2. It is found that the F1 of adding any separate structural feature is higher than that of using the base pairing feature alone, and lower than that of the combination of the base pairing feature and four structural features. This shows that our structural features can well compensate for the reduced performance caused by the sparsity of the base pairing feature.

To explore the generalization performance of these features, we also test them on the TestSetA_112 and TestSetB_112 datasets. We find that these base pairing signature (BPS) and structural signature (SS) features have good generalization results. The F1 index by the base pairing feature is generalized from 0.790 in TestSetA_112 to 0.328 in TestSetB_112, while the result by the structural feature is changed from 0.786 in TestSetA_112 to 0.321 in TestSetB_112, with similar generalization capabilities. Additionally, the combination of these two features achieves better generalization performance, from 0.816 to 0.358 in the F1 index, with a relatively 9.1% improvement on the base pairing feature alone and a relatively 11.5% improvement on the structural feature alone on TestSetB_112. The use of all features has a relatively 12.8% improvement in generalization performance compared to the base pairing signature.

### 3.3. SWIN-RNA Pre-Training Ablation

We conduct the ablation experiment on random weights and pre-trained weights of SWIN-RNA on the RSS prediction and RNA classification tasks, and compare the performance by fine-tuning two kinds of weights, as shown in Table 3. We find that the result by pre-trained weights surpasses that by random weights in all indicators. In addition, we visualize the feature embedding result on the nRC dataset under two weights and the feature embedding after fine-tuning the pre-trained weights, as shown in Figure 2. We find that the feature embedding of different classes presents a large random distribution by the random initialization, while a few categories have been clustered together through the pre-trained weights, such as ribozyme, 5S_rRNA, and 5_8S_rRNA.

In addition, we plot the local average entropy and local average attention distance of the attention heads at different layers in SWIN-RNA, as shown in Figure 3. We compare the results using weights by the pre-trained model and random initialization.

The average entropy by the pre-trained model is smaller than that of the random initialization both locally and globally, indicating that the attention area is more focused (smaller entropy), and the attention of the pre-trained model is more local (smaller average distance). Namely, SWIN-RNA has the lower attention distance and entropy compared with the random initialization. At the same time, we notice that the distance and entropy of different attention heads are distributed in a large range, which can pay attention to different features and have a larger attention range, making the model have a more focus on local and global tokens, with a wide and concentrated accumulation.

### 3.4. Multimodal Ablation

We conduct the ablation experiment on the single sequence model, single vision model and their combined multimodality. In the protein–RNA interaction prediction task, we select several datasets (CPSF3 in the HEK293 lineage, CDC40 in the HepG2 lineage, and ATXN2 in the HEK293T lineage), in which there are poor prediction results by the PrismNet model. By applying our DRFormer, the prediction performance improves significantly compared to using either a single sequence modality or a vision modality alone, as shown in Table 4. At the same time, we find that there are different performances by DRFormer with different numbers of multimodal fusion block layers on these datasets. Overall, a DRFormer model with two multimodal fusion block layers performs the optimized prediction, and for most evaluation indexes, DRFormer outperforms PrismNet and DNABERT methods. Especially, in terms of the F1 index on the CDC40_HepG2 dataset, SWIN-RNA has been better than DNABERT, indicating that our pre-training model can effectively extract features.

### 3.5. RNA Downstream Task Analysis

#### 3.5.1. RSS Prediction

For the RSS prediction, we select UFold [13] (model based on image vision), RNAErnie [9] (a relatively new large model), MXfold2 [8] (a method based on thermodynamics), and SPOT-RNA [26] (transfer learning–based model) for the comparison. The comparison results are shown in Table 5.

On the TS0_112 dataset, DRFormer outperforms above four models. Note that SPOT-RNA trains on the complete TR0 and PDB datasets, and then uses a multimodal fusion method for prediction, so it achieves the relatively good results on the corresponding TS0 subset. However, our DRFormer is only trained on TR0_112 with less training data, and achieves better results than SPOT-RNA, which shows that our model can effectively capture the corresponding structural information. In addition, it is found that our model also achieves the best results in F1 and precision on the TS0_112 dataset, leading the second SPOT-RNA by about 1% in F1 and about 10.8% in precision.

For SPOT-RNA and DRFormer, we also visualize some portions of predicted structures for analysis in Figure 4 (Section B.1 Figure A1 for more results). In Figure 4a, our model predicts more correct structures than SPOT-RNA (the green box is the correct region, and our model predicted four regions), which shows that our predicted structures are more accurate than SPOT-RNA. Although SPOT-RNA tries to predict some other structures, most of them are wrong as shown in the red boxes. In Figure 4b, our model only predicts one incorrect structure, while the prediction results of SPOT-RNA have a relatively higher error rate, and SPOT-RNA only predicts one structure correctly, which shows that the structure predicted by our model is more credible.

In addition, we do the motif analysis of subsequences existing paired relationship on the TS0_112 dataset, as shown in Figure 5. In this figure, the left shows the motif with paired bases in the real structure, and the right shows the motif with paired bases by our prediction model. We find that the predicted motif result is almost same the real result, which shows that our prediction model can precisely obtain the structural information.

#### 3.5.2. RNA Sequence Classification

In this section, we evaluate our model for the RNA sequence classification, and choose the commonly used RNAcon [27], nRC [21], ncRFP [28], RNAGCN [29], ncRDeep [30], and several large models such as RNABERT [31], RNAErnie [9], and RNA-MSM [32] for comparison.

The comparison results are shown in the left of Figure 6 (Section B.2 Table A1). In the RNA sequence classification, our DRFormer achieves the best results compared with above comparative models, followed by RNAErnie. Note that RNAErnie has added RNA type as a special IND label (a special token like CLS token) for pre-training, and it achieves good results when performing RNA sequence classification on the nRC dataset. Our multimodal model DRFormer effectively combines sequence and structural information to obtain the best result of 0.948, which is about 1.12% ahead of the second RNAErnie. The right of Figure 6 shows the classification confusion matrix of our model. We can see that our model can make satisfactory predictions for most categories of RNA sequences, such as 5S_rRNA, Intron_gpll, tRNA, etc., with an accuracy rate of more than 95%.

#### 3.5.3. Protein–RNA Interaction Prediction

In the protein–RNA interaction prediction, we select PrismNet and PrismNet_Str [12], GraphProt [33], BERT-RBP [34], and RNAErnie [9] for comparison. The comparison results are shown in Figure 7 and Figure 8 (Section B.3 Table A2). We observe that PrismNet_Str introduces additional icSHAPE structural information compared to PrismNet, and it indeed brings a significant improvement, indicating that structural feature can provide useful information for prediction. Of course, as the first large model proposed in the protein–RNA interaction prediction task, BERT-RBP shows a good indicator of MCC, far exceeding those machine learning models and non-large deep learning models. Note that the MCC of RNAErnie on some RBPs (RNA-binding proteins) is always 0, which may be influenced by the instability of the model. At the same time, the performance of RNAErnie is not as good as BERT-RBP. Although RNAErnie is leading in the F1 index, it may be the result of RNAErnie oscillation, as the MCC for the remaining is 0 two times.

Our multimodal model applies a variety of structural feature and also uses a sequence module to combine sequence information. Therefore, it is better than the second-top BERT-RBP model based on the pre-trained DNABERT in terms of MCC, AUC, F1, and other indicators. And compared with PrismNet, PrismNet_Str, GraphProt, and RNAErnie, our model is significantly in the lead in most RBP performance indices.

In addition, we visualize the attention analysis and the actual structure distribution on the RNA sequence as the case study in Figure 9. It illustrates a total of three RNA sequence fragments, in which A and B are different RNA fragments corresponding to SND1 in the K562 lineage, and C is the RNA fragment corresponding to AUH in the K562 lineage. Through the analysis of the first and second fragments, we find that our model does not express well (the green box part) on the single-stranded structure (icSHAPE score tends to 1), and would prefer double-stranded structures (icSHAPE score tends to 0). This is consistent with what we know about the preference of SND1, which prefers double-stranded structures [12].

At the same time, we also analyze some motifs in the mCross database [35]. As shown in Figure 9D, the motif of SND1 in the K562 lineage has a certain significance in the GGCC sequence segment (red box in Figure 9A,B). Figure 9E shows the motif of AUH in the K562 lineage. It is also significant in UAUC, which is consistent with our high attention expression area (red box in Figure 9C), indicating that our model can pay good attention to the corresponding sequence segment.

### 3.6. Generalizability Analysis—DNA Downstream Task Analysis

In order to verify the generalization performance of our DRFormer model, we also analyze its applicability on the DNA downstream tasks. We conduct the experiment on some parts of GUE datasets. Specifically, we refer to the 13 datasets of Zhou et al. [9] and analyze three tasks of Transcription Factor Prediction (Human), Core Promoter Detection (Human), and Transcription Factor Prediction (Mouse). At the same time, we compare it with the current large models based on sequence text, such as DNABERT, Nucleotide Transformer (NT) [36], and RNAErnie [9]. We find that our model performs better on most datasets, which is shown in Figure 10 (Section B.4 Table A3, Table A4 and Table A5).

For the long sequences, we first adopt the following strategy for treatment: we choose the first 112 bases of the original sequence for model training and prediction, and then select 13 datasets from the remaining GUE dataset [9] and compare two tasks of Epigenetic Marks Prediction (EMP) and Promoter Detection (PD) for analysis. The overall performance is shown in Table 6 (Section B.4 Table A6 for detailed results). We find that DRFormer leads in 11 of 13 datasets, with an average score of 0.601, 4.34% ahead of the second method. This shows that even if our model only uses part of the sequence for training, our model still works well. Of course, for the complete long sequences, we can also employ the complete sequence for training and prediction. We conduct the experiment for PD task on the tata dataset. After training through the complete sequence, we find our MCC is 0.798, which is about 5.7% higher than the sequence truncation strategy.

## 4. Discussion

This paper proposes a novel benchmark model DRFormer for the analysis of various downstream tasks of RNA sequences. The main contributions of this model are as follows: (1) in terms of structural feature extraction, we introduce a new longest possible secondary structure, sequence repeat structure, base missing structure, and spatial distance feature based on the existing sequence pairing feature, Manhattan distance feature, and RNA coding matrix, and fully exploit the structural and distance features and propose a more comprehensive structural vision feature; and (2) we pre-train the structural vision feature and combine it with the sequence feature to construct a multimodal model. The self-attention module and the cross-attention module fully integrate two-modal features for various downstream analysis.

At present, few studies have combined the vision feature with RNA structural information and the sequence feature to build the multimodal model. Our DRFormer model integrates RNA sequence module and structural vision module in a multimodal way, bringing excellent performance and generalization potential to downstream RNA tasks. For example, it outperforms existing advanced technologies in RNA downstream tasks, such as RNA secondary structure prediction, protein–RNA interaction prediction, and RNA classification. At the same time, it also achieves satisfactory results in generalized DNA downstream experiments.

Although the proposed DRFormer has achieved the satisfactory performance on various RNA downstream analyses, there are still some problems to be improved in future. For example, in RNA downstream tasks such as the sequence classification, protein–RNA interaction prediction, and RNA secondary structure prediction, the entire sequence can be input into our model and the prediction is slightly better than our current result. But the computational complexity of our model will significantly increase. Therefore, the sequence length is limited to 112, considering the great equipment and time requirement. How to combine the overlapping sequence windows and hierarchical strategies with our current model to improve the prediction performance will be our future work. In addition, when using the SWIN-RNA pre-trained model for calculation, this model can expand the scaling range through the Patch Embed (PE) layer, which can speed up the model training, but this would cause some kind of accuracy impact. At the same time, the structural features that we designed are implemented based on the standard pairing method [22], which may miss some pseudo-structures (the cross-pairing base pair information in the pseudoknot or the contribution of atypical pairing regions), thereby affecting the acquisition of structure and the performance of the model. In addition, we find that the longest possible secondary structure, sequence repeat structure, base missing structure, and spatial distance feature have been verified to be effective in RNA sequence analysis. How to expand these new structural and distance features and apply them to other sequence analyses will also be our future work.

## Figures and Tables

**Figure 1 genes-16-00284-f001:**
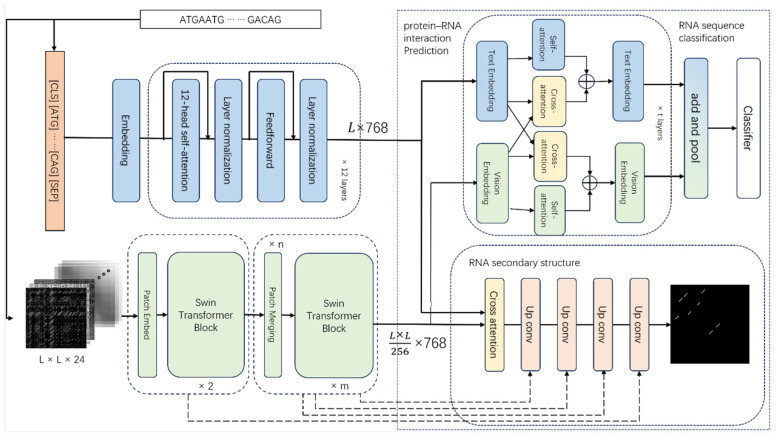
Overall framework of the model.

**Figure 2 genes-16-00284-f002:**
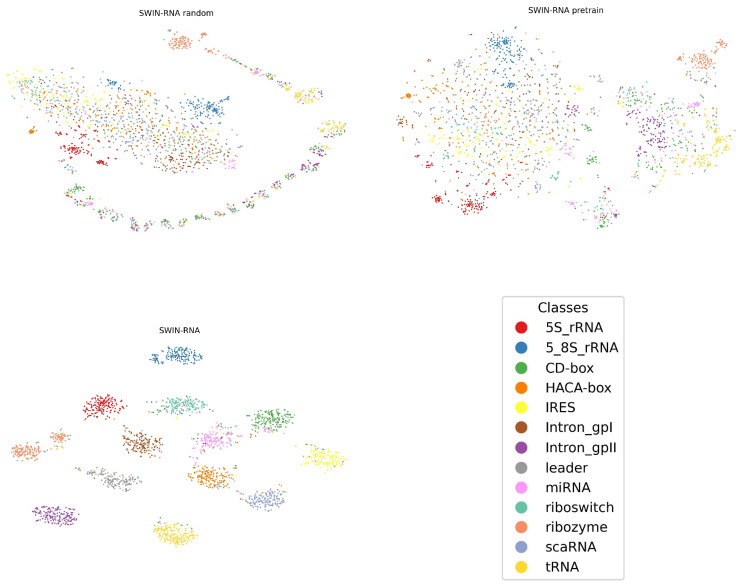
Feature embedding visualization on the nRC dataset.

**Figure 3 genes-16-00284-f003:**
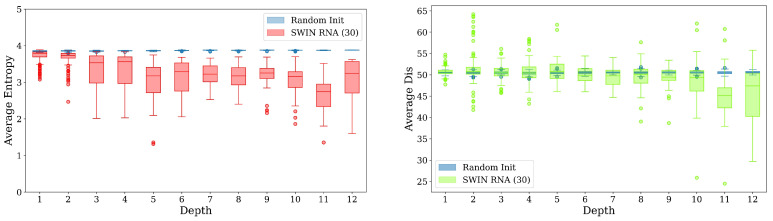
Attention distance and entropy analysis. (**Left**): Local average entropy. (**Right**): Local average attention distance.

**Figure 4 genes-16-00284-f004:**
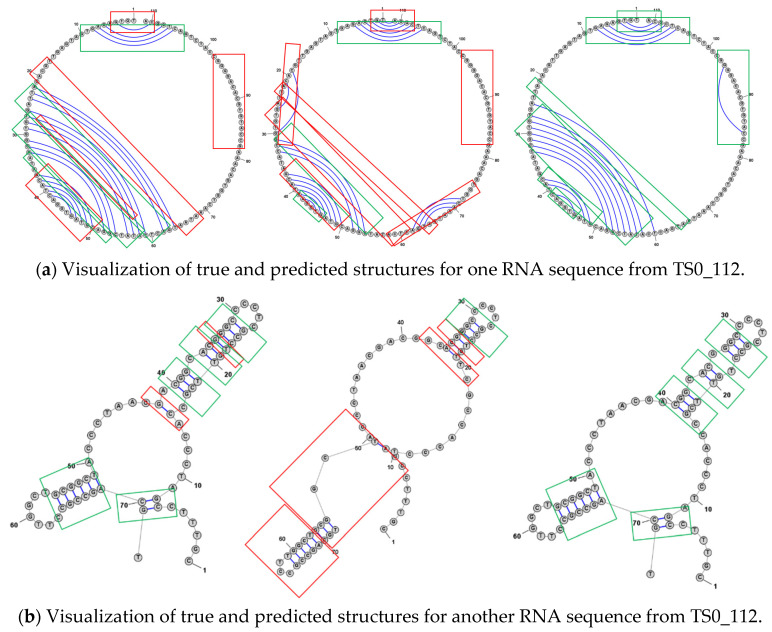
Comparison of true structures and predicted structures by DRFormer and SPOT-RNA. (**Left**): DRFormer predicted structure. (**Middle**): SPOT-RNA predicted structure. (**Right**): True structure. In the figure, red boxes indicate prediction errors or no predictions, and green boxes indicate true structures.

**Figure 5 genes-16-00284-f005:**
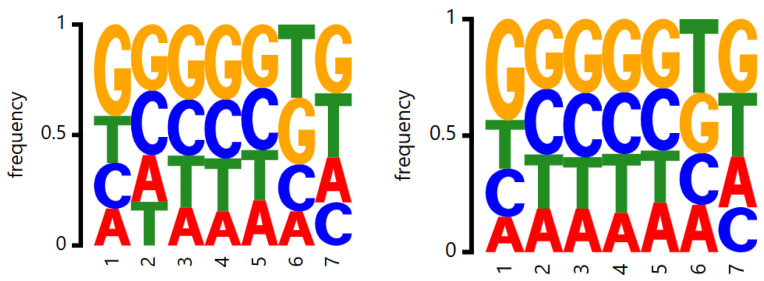
Motif analysis of subsequences existing paired relationship in the TS0_112 dataset.

**Figure 6 genes-16-00284-f006:**
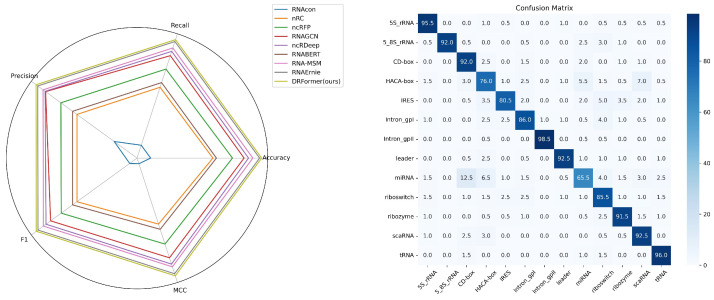
RNA sequence classification performance comparison.

**Figure 7 genes-16-00284-f007:**
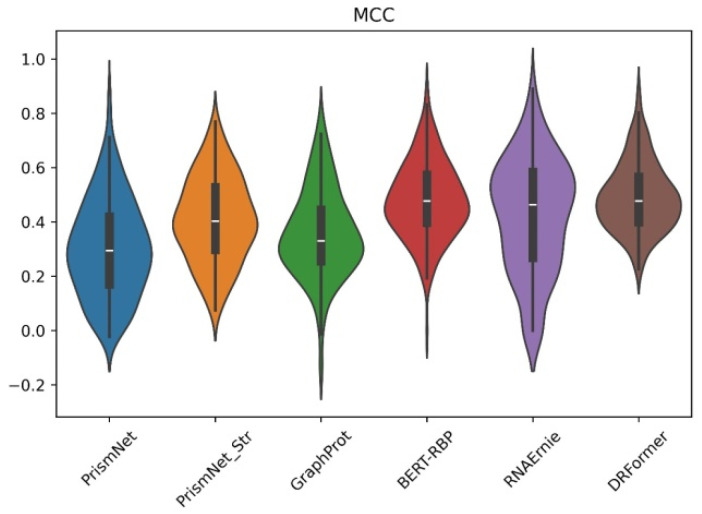
RBPs violin plot of MCC for different methods.

**Figure 8 genes-16-00284-f008:**
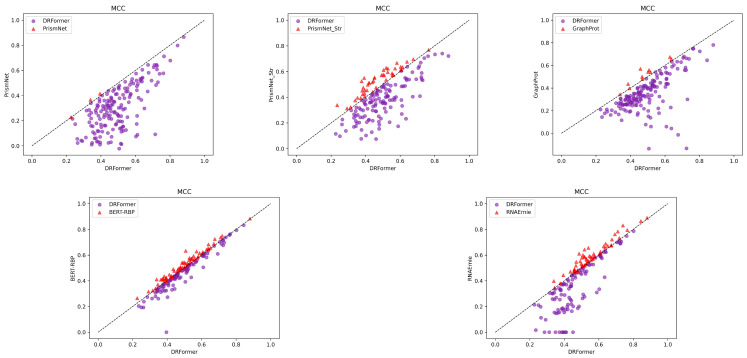
Performance comparison of DRFormer with PrismNet, PrismNet_Str, GraphProt, and BERT-RBP using multiple metrics. Each dot represents the corresponding metric score of DRFormer and the corresponding baseline model trained using the same RBP dataset, and the diagonal dotted line indicates that the performance of two models is the same. The dots in the figure represent DRFormer, and the triangles represent the corresponding baseline models.

**Figure 9 genes-16-00284-f009:**
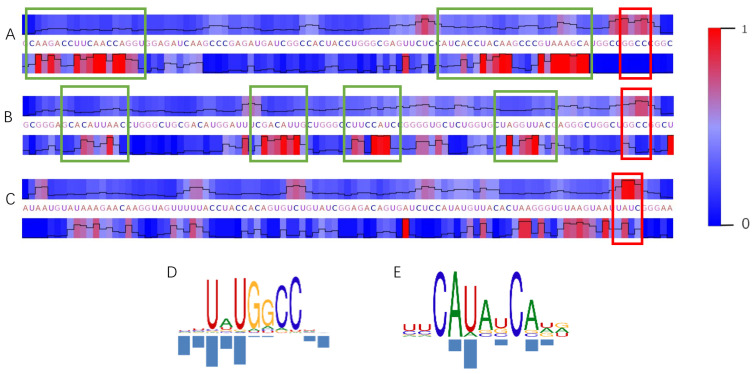
Attention analysis of DRFormer on the RNA sequence and its sequence structure. (**A**,**B**) represent different RNA fragments corresponding to SND1 in the K562 lineage, and (**C**) represents the RNA fragment corresponding to AUH in the K562 lineage. For any fragment, the (**top**) represents the attention area of our model, the (**middle**) represents the actual sequence, and the (**bottom**) represents the value corresponding to icSHAPE, where 1 represents single-stranded, and 0 represents double-stranded. (**D**) The motif of SND1 in the K562 lineage in mCross. (**E**) The motif of AUH in the K562 lineage in mCross.

**Figure 10 genes-16-00284-f010:**
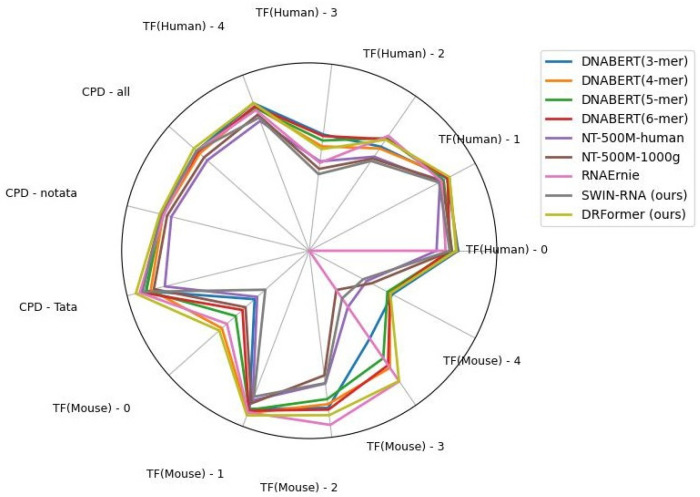
Performance chart for TF (human), CPD, and TF (mouse).

**Table 1 genes-16-00284-t001:** Feature block ablation.

	TS0_112			TestSetA_112			TestSetB_112		
	F1	SEN	PPV	F1	SEN	PPV	F1	SEN	PPV
BPS	0.587	0.593	0.624	0.790	0.781	0.818	0.328	0.302	0.372
SS	0.613	0.618	0.664	0.786	0.767	0.836	0.321	0.264	0.452
BPS + SS	0.631	0.640	0.672	0.816	0.817	0.833	0.358	0.325	0.422
BPS + DIS + SS	0.639	0.639	0.689	0.800	0.804	0.810	0.370	0.333	0.438

**Table 2 genes-16-00284-t002:** Structural feature ablation.

	TS0_112			TestSetA_112			TestSetB_112		
	F1	SEN	PPV	F1	SEN	PPV	F1	SEN	PPV
BPS + LPSS	0.629	0.650	0.657	0.813	0.807	0.833	0.356	0.327	0.410
BPS + RCM	0.621	0.624	0.670	0.810	0.781	0.869	0.332	0.285	0.437
BPS + BMS	0.601	0.596	0.656	0.797	0.781	0.831	0.314	0.291	0.355
BPS + SRS	0.602	0.603	0.658	0.794	0.793	0.809	0.329	0.305	0.378

**Table 3 genes-16-00284-t003:** SWIN-RNA weight ablation experiment.

	RSS									RNA Classification		
	TS0_112			TestSetA_112			TestSetB_112			nRC		
	F1	SEN	PPV	F1	SEN	PPV	F1	SEN	PPV	MCC	F1	ACC
Random Init.	0.639	0.639	0.689	0.800	0.804	0.810	0.370	0.333	0.438	0.870	0.879	0.880
Pre-train	0.683	0.707	0.711	0.826	0.812	0.863	0.419	0.357	0.584	0.896	0.904	0.904

**Table 4 genes-16-00284-t004:** Multimodal ablation experiment.

	RBPs								
	CPSF3_HEK293			CDC40_HepG2			ATXN2_HEK293T		
	MCC	AUC	F1	MCC	AUC	F1	MCC	AUC	F1
PrismNet	0.149	0.709	0.514	0.180	0.700	0.522	0.061	0.758	0.500
DNABERT (3-mer)	0.356	0.731	0.676	0.332	0.745	0.637	0.384	0.761	0.692
SWIN-RNA	0.315	0.723	0.650	0.280	0.710	0.638	0.279	0.727	0.619
DRFormer (1-layer)	0.358	0.756	0.679	0.328	0.739	0.640	0.406	0.777	0.702
DRFormer (2-layer)	0.382	0.764	0.691	0.358	0.733	0.672	0.376	0.765	0.680
DRFormer (3-layer)	0.353	0.749	0.675	0.346	0.738	0.672	0.347	0.762	0.647

**Table 5 genes-16-00284-t005:** RNA secondary structure performance table.

	TS0_112			ArchiveII		
	F1	SEN	PPV	F1	SEN	PPV
UFold	0.647	0.751	0.602	0.896	0.881	0.922
RNAErnie	0.574	0.686	0.519	0.930	0.938	0.925
MXfold2	0.629	0.731	0.582	0.895	0.907	0.885
SPOT-RNA	0.683	0.804	0.620	0.799	0.881	0.739
DRFormer (ours)	0.690	0.731	0.687	0.928	0.931	0.931

**Table 6 genes-16-00284-t006:** Overall performance of EMP and PD300 tasks.

Model	Num. Top 1	EMP Ave. Scores	PD300 Ave. Scores	Ave. Scores
DNABERT (3-mer)	1	0.496	0.846	0.576
NT-500M-human	0	0.453	0.855	0.546
NT-500M-1000g	1	0.477	0.866	0.567
DRFormer (ours)	11	0.522	0.864	0.601

## Data Availability

DRFormer (including data and code) is publicly available at https://github.com/llfzllfz/DRFormer (accessed on 4 January 2025).

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
