# Peer review of "DRFormer: A Benchmark Model for RNA Sequence Downstream Tasks"

_genes, 2025, doi:10.3390/genes16030284_

Round 1

Reviewer 1 Report

Comments and Suggestions for Authors

Summary: The paper proposes DRFormer, a benchmark model for RNA sequence downstream tasks. DRFormer leverages structural vision features and sequence features to construct a multimodal model. It utilizes the SWIN-RNA submodel pre-trained on RNA secondary structure and integrates it with a sequence model using cross-attention mechanisms. DRFormer demonstrates state-of-the-art performance in various tasks, including RNA sequence classification, protein–RNA interaction prediction, and RNA secondary structure prediction. Additionally, it generalizes well to DNA downstream tasks.

Strength: This is a very interesting paper combining vision and sequence features using cross-attention, enhancing feature extraction and improving model performance. From machine learning point of view, applying cross-attention and feature extraction for RNA sequence processing is efficient. This novel design is the major merit of this paper, which is confirmed by the new state-of-the-art performances. Beyond that, the paper is well-written, well organized and with appropriate citations. English expression is okay in this paper.

Weakness: The model is limited to sequences of length 112 for RNA secondary structure prediction. Will this impact the ability to process long RNA sequences? If not, please ignore this comment.

What is the computational complexity of the proposed model? In general attention-mechanism should be computationally efficient. But it will provide a more comprehensive evaluation to the model is the computational complexity is discussed briefly.

Conclusion: In general, from machine learning point of view, this is a good paper applying efficient attention mechanism to seq learning with new state-of-the-art results. I will recommend to accept after minor revision. Please take care of the above two weaknesses.

Reviewer 2 Report

Comments and Suggestions for Authors

This paper addresses a critical challenge in RNA research, which is constructing effective benchmark models for downstream RNA analysis. The use of vision-based features from RNA secondary structure and sequence distance is an interesting approach, leveraging recent advancements in transformer models. The combination of SWIN-RNA (vision-based model) and DNABERT (sequence model) is innovative and aligns with deep learning advancements in genomics. The model is evaluated on diverse, well-established datasets (e.g., NONCODEV6, bpRNA-1m, CLIP, nRC, GUE) and achieves state-of-the-art results in multiple RNA-related tasks, showcasing its robustness and effectiveness. Although, the paper has several merits, I have some of the following concerns to be addressed in the manuscript:

  1. The paper mentions that sequences longer than 112 bases are trimmed, leading to some performance loss. Could overlapping sequence windows or hierarchical attention mechanisms mitigate this issue?
  2. The structural prediction analysis focuses on accuracy but does not discuss biological insights gained from DRFormer’s predictions. Does DRFormer reveal novel RNA motifs or structure-function relationships? A brief analysis of biological relevance would enhance the impact of the study.
  3. Including a case study of an RNA sequence where DRFormer made a unique or biologically relevant predictionwould increase the paper’s impact for experimental biologists.                                                                                                                                    
Comments on the Quality of English Language

Some of the minor concerns are regarding the English language/ scientific writing of the manuscript are as follows:

  1. For instance the sentences such as "After using our DRFormer, the prediction effect is improved compared with single sequence modality and vision modality" can be rephrased for more clarity.
  2. Terms such as “prediction effect” should be replaced with more standard phrases like “prediction accuracy” or “performance”.
  3. Incorporation of appropritate grammatical changes are necessary. (Example: "We conduct ablation experiment ..." should be written as "We conduct an ablation experiment ...") 
